# GPs' attitudes, beliefs and behaviours regarding exercise for chronic knee pain: a questionnaire survey

Elizabeth Cottrell,[1] Nadine E Foster,[2] Mark Porcheret,[1] Trishna Rathod,[1,2] Edward Roddy[1]

► Prepublication history and additional material is available online. To view please visit the journal online (http://dx.doi.org/ 10.1136/bmjopen-2016-014999).

[1]Arthritis Research UK Primary Care Centre, Research Institute for Primary Care and Health Sciences, Keele University, UK
[2]Keele Clinical Trials Unit, David Weatherall Building, Keele University, Staffordshire, UK

**Correspondence to**
Dr Elizabeth Cottrell;
e.cottrell@keele.ac.uk

## ABSTRACT

**Objectives** The aim of this study was to investigate general practitioners' (GPs) attitudes, beliefs and behaviours regarding the use of exercise for patients with chronic knee pain (CKP) attributable to osteoarthritis.

**Setting** Primary care GPs in the UK.

**Participants** 5000 GPs, randomly selected from Binley's database, were mailed a cross-sectional questionnaire survey.

**Outcome measures** GPs' attitudes and beliefs were investigated using attitude statements, and reported behaviours were identified using vignette-based questions. GPs were invited to report barriers experienced when initiating exercise with patients with CKP

**Results** 835 (17%) GPs responded. Overall, GPs were positive about general exercise for CKP. 729 (87%) reported using exercise, of which, 538 (74%) reported that they would use both general and local (lower limb) exercises. However, only 92 (11% of all responding) GPs reported initiating exercise in ways aligning with best-evidence recommendations. 815 (98%) GPs reported barriers in using exercise for patients with CKP, most commonly, insufficient time in consultations (n=419; 51%) and insufficient expertise (n=337; 41%).

**Conclusions** While GPs' attitudes and beliefs regarding exercise for CKP were generally positive, initiation of exercise was often poorly aligned with current recommendations, and barriers and uncertainties were reported. GPs' use of exercise may be improved by addressing the key barriers of time and expertise, by developing a pragmatic approach that supports GPs to initiate individualised exercise, and/or by other professionals taking on this role.

## Strengths and limitations of this study

► This large UK survey is the first known to directly, specifically and concurrently investigate the attitudes, beliefs and behaviours of general practitioners (GPs) regarding exercise for patients with chronic knee pain (CKP). Exercise initiation did not concur with best-evidence recommendations and GPs reported uncertainties and barriers in relation to using exercise.

► The questionnaire was pretested and piloted before being used in this main study. Use of a vignette to investigate clinical management ensured a consistent patient scenario to all GPs and minimised the confounders inherent in observational research using real patients.

► Limitations of this study include the likely overestimation of exercise use among GPs given the low response rate (response bias), the self-report nature of the questionnaire (social desirability bias), inability to explore underlying reasons for responses and the relatively uncomplicated vignette case. However, as GPs should be using exercise for all patients with CKP, the results of this survey are valuable for indicating an apparent evidence-practice gap in the way in which GPs employ exercise with this patient group.

## INTRODUCTION

General practitioners (GPs) are the most frequently accessed source of formal medical advice and treatment for patients in the UK with chronic knee pain (CKP).[1–3] CKP is defined in this study as being synonymous with clinical knee osteoarthritis (OA),[4] that is, mechanical knee pain, with or without loss of function, and with or without radiographic changes consistent with OA, that has lasted for at least 3 months in people aged 45 years and older,[5] and for which an alternative diagnosis is unlikely.[4] Globally, OA is among the leading causes of years lived with disability,[6] thus it is unsurprising that CKP is a common presentation to GPs.[7 8] Exercise, comprising both local (lower limb focused) and general (aerobic) exercise, is recommended as core treatment for CKP,[4 9] its provision is one of the eight UK OA quality standards,[10] and international OA experts recognise provision of information about regular physical activity and individualised exercise to patients as essential.[11] Empirical research evidence now unequivocally demonstrates that general aerobic, local strengthening and flexibility exercises improve pain and function in patients with CKP.[12] In line with wider self-management strategies, best practice outlined by the

National Institute for Health and Care Excellence (NICE) OA guidelines with regards to integrating exercise into the management of patients with CKP involves providing verbal advice about both general and local exercise (which should be specific and individualised[4 13]) supported with written information.[4] Where GPs feel unable to provide specific and individualised advice, referral of patients to appropriate exercise specialists (eg, physiotherapist) would be appropriate. While it is recognised that delivery of care for CKP is multidisciplinary, the exact roles and explicit expectations of GPs (and other professionals) regarding the delivery of core management approaches is not provided within current guidelines. This could have the consequence that no professional undertakes certain activities in the belief that others will.

To maximise patient outcomes, GPs should align their management with best-evidence recommendations. As sociocognitive behavioural theories suggest an association between individuals' attitudes and beliefs and their behaviours,[14–16] concurrent investigation of attitudes, beliefs and behaviours of GPs was undertaken. A systematic review revealed a paucity of data specifically examining GPs' use of exercise for patients with CKP, however attitudes regarding exercise were variable, it appeared to be underused and its implementation by GPs was unclear.[5] The role that GPs perceive themselves to have in delivering these management approaches was also not clear. The aim of this cross-sectional questionnaire survey was to identify the attitudes, beliefs and behaviours of UK GPs regarding the use of exercise for patients with CKP. Analysis of factors associated with the use of exercise among this group have been published elsewhere.[17]

## METHODS

A cross-sectional survey was used to investigate GPs' attitudes, beliefs and behaviours regarding exercise for CKP. The survey tool had previously been developed through pretesting by a local group of GPs and a subsequent pilot study with 172 UK GPs,[18] which was designed to investigate the likely response to the questionnaire, to finalise the survey tool and to test methods to maximise quantity and quality of responses.

In this main study, GPs were mailed an eight-page questionnaire (see online supplementary file 1), a cover letter and a postage-paid reply envelope in January 2014. Non-responders were sent a reminder postcard after 2 weeks and, 4 weeks after the initial mailing, persistent non-responders were mailed a second copy of the questionnaire with a cover letter and postage-paid reply envelope. At each stage non-responders who did not wish to complete the full questionnaire were invited to provide minimum data sets (MDS; gender, year of qualification, practice size and setting) and a reason for non-response. Attitude statements associated with a 5-point Likert Scale explored GPs' attitudes and beliefs about exercise for CKP. These were minimally adapted from the work of Holden *et al* who investigated this among physiotherapists[19] and

**Table 1** Vignette used in the questionnaire to assess GPs' reported behaviours

| Patient | Mrs Jones, 58-year-old prison officer |
| --- | --- |
| History | First presentation of gradually worsening bilateral knee pain (right worse than left) over 2 years. No history of trauma. Pain always present when walking and at rest, worst when climbing stairs. No night pain. Managing activities of daily living. Difficulty gardening. Stopped going to gym—thinks was making pain worse. Only treatment tried is ibuprofen once or twice when pain 'really bad'—no benefit. Came today finding work increasingly difficult due to the stairs. Usually well—no comorbidities. |
| Medication | Nil. |
| Examination | Body mass index 33. Knees—bilaterally no effusions. Joint tenderness on palpation. Bilateral coarse crepitations. Slightly reduced flexion of the right knee. Hips—no abnormality detected. |

older adults with CKP[20] and were derived from the MOVE consensus recommendations, designed to help healthcare professionals (HCPs) to initiate exercise in the management of a patient with lower limb OA.[21] GPs' reported clinical behaviour was investigated using multiple response questions associated with a vignette case (see table 1). GPs reporting to use exercise were requested to indicate the type of exercise and how this was initiated. A multiple response item, with space for free text, investigated GPs' experiences of barriers to using exercise for CKP. Completion and return of the questionnaire by the GP was interpreted as consent to participate in the study.

A minimum sample size of 288 responding GPs was required to estimate the use of exercise, based on a conservative estimate of 75% reporting exercise use informed by the pilot study[18] and a margin of error of <5%.[22] After increasing the minimum sample size to adjust for other planned regression analyses (exploring associations between reported exercise use and attitudes/beliefs (published elsewhere[17])) and anticipating a response rate of 20%,[18] 5000 UK GPs were randomly selected from Binley's database, a database containing the contact details of GPs in the UK which is updated quarterly. Binley's extracted a simple random sample of GPs from their database and removed and replaced any GPs included in the sample used for the previous pilot study. Study exclusion criteria were not being a GP and not having managed someone with CKP in the previous 6 months.

GPs' practice postcodes from each UK country were transformed into their corresponding Index of Multiple

Deprivation rank[23–26] and split into quintiles (1=most deprived, 5=least deprived). Responses to the GP attitude statements were condensed into three categories ((strongly) disagree, neither disagree nor agree, and (strongly) agree) and free-text responses (associated 'other' responses and regarding type of exercise the GPs would use) underwent thematic analysis, categorising responses into predefined categories that emerged from responses to the pilot study and developing new categories as appropriate, before commencing descriptive analyses. Responses to GP attitude statements were interpreted as follows: unanimity=100%, consensus=75%–99%, majority view=51–74%, no consensus=0%–50%.[19 27] To assess for possible response bias in questionnaire respondents versus MDS responders, demographic data of each type of responders were compared using logistic regression to obtain unadjusted OR with 95% CI (CI; gender, practice area deprivation and practice type) and mean difference with 95% CI (mean years since qualification and mean number of GPs in respondents practice). An a posteriori analysis was undertaken due to the timing of the main survey coinciding with publication of the revised version of the NICE OA guidelines on 12 February 2014 (4 weeks after the baseline mailing of the survey). To establish whether the publication of these guidelines, and the publicity associated with this event had an impact on the proportion of GPs using exercise, logistic regression was used to compare the use of exercise among responses received before the date of publication of the revised guideline with those responses received after. All analyses were performed using IBM SPSS Statistics (V.20).

## RESULTS

### Response

Of the 5000 GPs that were sent the questionnaire, 58 responders met one or more exclusion criteria and 835 returned a completed questionnaire (adjusted response 17%). A further 470 provided MDS. The most common reason for returning MDS, rather than a full response was, having insufficient time (n=408, 87%). The characteristics of GPs responding with an MDS were similar to those responding with a full questionnaire, except that they had been qualified for longer and were more likely to work in practices in the most deprived areas (table 2). When compared with GPs with practice postcodes in the

**Table 2** Demographic details of questionnaire respondents versus those providing MDS

| | | Response type | | |
| | | **MDS** **(n=470)** | **Completed questionnaire (n=835)** | **OR/mean difference (95% CI)** |
| **Variable** | **Category** | | | |
|---|---|---|---|---|
| Gender | Male | 247 (53%) | 401 (49%) | OR 1.00 |
| | Female | 219 (47%) | 417 (51%) | OR 1.17 (0.93 to 1.47) |
| Practice area deprivation | Most deprived | 121 (26%) | 181 (22%) | **OR 0.63 (0.45 to 0.89)** |
| | Second most deprived | 106 (23%) | 156 (19%) | **OR 0.62 (0.44 to 0.88)** |
| | Mid-deprived | 85 (18%) | 202 (24%) | OR 1.00 |
| | Second least deprived | 84 (18%) | 160 (19%) | OR 0.80 (0.56 to 1.16) |
| | Least deprived | 73 (16%) | 135 (16%) | OR 0.78 (0.53 to 1.14) |
| Practice type | Urban | 254 (56%) | 449 (54%) | OR 1.00 |
| | Semirural | 155 (34%) | 275 (33%) | OR 1.00 (0.78 to 1.29) |
| | Rural | 43 (10%) | 103 (13%) | OR 1.36 (0.92 to 2.00) |
| Mean (SD) years since qualification | | 21.64 (10.03) | 18.40 (10.33) | Mean difference = −3.24 (-2.06 to- 4.42) |
| Mean (SD) number of GPs in respondent's practice | | 6.44 (3.67) | 6.44 (3.20) | Mean difference = <0.01 (−0.38 to 0.39) |
| **Information only requested in questionnaire** | | | | |
| Type of GP | GP partner | – | 656 (79%) | – |
| | Salaried GP | – | 151 (18%) | |
| | Locum GP | – | 20 (2%) | |
| | Other | – | 5 (1%) | |
| GP with special interest in musculoskeletal conditions | | – | 50 (6%) | – |
| Received postgraduate education about CKP | | – | 319 (39%) | – |
| Personal experience of CKP | | – | 166 (20%) | – |

Maximum missing data for any cell were 6%.
CKP, chronic knee pain; GP, general practitioner; MDS, minimum data set.

mid-deprived quintile (OR (95% CI)), those in the most deprived (0.72 (0.60 to 0.87)) and second most deprived (0.76 (0.62 to 0.92)) were significantly less likely to respond in any way (completed questionnaire or MDS), although the absolute difference in the proportions responding were small.

### Attitudes of GPs regarding exercise for CKP

Table 3 summarises responses to GP exercise attitude statements, none were unanimous. Generally, GPs were more positive about general exercise than local exercise, particularly with regards to safety and efficacy. However, more GPs agreed that increasing the strength of the muscles around the knee stops the knee problem getting worse compared with those who agreed that increasing overall physical activity would do the same. No consensus was reached regarding the statement 'exercise works just as well for everybody, regardless of the amount of pain they have', however the greatest proportion of respondents disagreed. GPs recognised the need to tailor exercises to individual patients, acknowledged the importance of adherence with exercise but placed responsibility for adherence on the patient.

### GPs' reported use of exercise for CKP

Of the 835 respondents, 729 (87%) reported using exercise of some type for the vignette case. Figure 1 summarises the types of exercise and initiation methods that GPs reported they would use. Among GPs reporting to suggest general exercise (n=347), the most common recommendations were swimming (49%), walking (41%) and cycling (34%). Only 17 (5%) GPs explicitly stated that general exercise should be tailored to patient's abilities and/or interests. Among GPs reporting to use exercise, 413 (57%) stated they would achieve this by referring the patient to a physiotherapist. Table 4 cross-tabulates the exercise initiation strategies GPs reported to use for both general and local exercise and shows the most common combinations of approaches were *suggesting* general exercise and *demonstrating* local exercise, and giving the patient a leaflet about both exercise types. Thirty-two (6%) GPs reporting to use both exercise types stated they would achieve this solely by referring the patient to another HCP. Ninety-two GPs (17% of those using both local and general exercise, 11% of all respondents) reported to use strategies aligned with evidence-based recommendations;[4] they advised, or referred for, local and general exercise and provided written information for both exercise types (table 4). The use of exercise was not significantly different among responses received after the publication of the revised NICE OA guidelines (273/314, 87%) when compared with those received before (456/521, 88%; OR 0.95 (95% CI 0.62 to 1.44)).

### Use of follow-up

Of the 729 GPs reporting to use of exercise, 494 (68%) stated that they would follow-up the vignette patient to establish ongoing engagement with regular exercise.

This was most commonly achieved through opportunistic follow-up (n=303, 61%) which most GPs (n=253, 84%) suggested would occur if the vignette patient failed to improve and reconsulted.

### Barriers to exercise use

Most GPs (n=815, 98%) reported having experienced barriers when using exercises for patients with CKP which included: (1) service-related, (2) GP-related and (3) (perceived) patient-related barriers (figure 2). The most frequently reported barriers were insufficient time (n=419, 51%), insufficient expertise (n=337, 41%) and the perception that patients prefer other management options (n=291, 36%).

## DISCUSSION
### Summary

This cross-sectional questionnaire survey sought to identify the attitudes, beliefs and behaviours of UK GPs regarding the use of exercise for patients with CKP. While most GPs agreed that they should 'prescribe' local and general exercise to all patients with CKP, believed that CKP is improved by local and general exercise, and reported that they would use exercise in the management of the vignette patient, only a tenth of responding GPs reported initiating exercise in a way that matches best-evidence recommendations.[4] This evidence-practice gap is perhaps unsurprising given that most GPs reported key barriers of time and expertise. A small number of GPs reported relying solely on referring the vignette patient to another HCP for both exercise types. This approach delays commencement of exercise as patients are not equipped to start exercising immediately after the consultation with the GP, relies on the receiving HCP delivering best practice and may not represent efficient use of services, if this is representative of GPs' usual approach for patients with CKP.

### Comparison with existing literature

Variable GP attitudes regarding exercise for CKP were recognised in an earlier systematic review[5] and the uncertainty regarding exercise efficacy identified by the current study supports this. GPs' responses to the MOVE attitude statements from the current study were compared with those from physiotherapists[19] and older adults with CKP.[20] In contrast to GPs being generally more positive regarding general exercise when compared with local exercise, physiotherapists were generally more positive about the safety and efficacy of local exercise,[19] and older adults with CKP reported low levels of agreement about the safety and efficacy of both types of exercise.[20] Although GPs were more positive about the efficacy and safety of both exercise types than either physiotherapists or those with CKP, the timing of the respective surveys must be considered. In the time between the previous and current surveys, new and/or revised versions of NICE,[4] Osteoarthritis Research Society International[9] and European League Against Rheumatism[28] guideline

**Table 3** Responses to GP attitude statements derived from the MOVE consensus recommendations[21]

| MOVE consensus proposition | Attitude statement | (Strongly) disagree | Neither disagree or agree | (Strongly) agree |
|---|---|---|---|---|
| **Items relating to the benefits of exercise (number of respondents)** | | | | |
| Prescription of both general (aerobic fitness training) and local (strengthening) exercises is an essential, core aspect of management for every patient with hip or knee OA | GPs should prescribe quadriceps strengthening exercises to every patient with CKP (n=822) | 8% | 22% | 69% |
| | GPs should prescribe general exercise, for example, walking or swimming, for every patient with CKP (n=824) | 3% | 8% | 89% |
| Both strengthening and aerobic exercise can reduce pain and improve function and health status in patients with knee and hip OA | Knee problems are improved by quadriceps strengthening exercises (n=824) | <1% | 11% | 88% |
| | Knee problems are improved by general exercise, for example, walking or swimming (n=824) | 1% | 7% | 93% |
| There are few contraindications to the prescription of strengthening or aerobic exercise in patients with hip or knee OA | Quadriceps strengthening exercises for the knee are safe for everybody to do (n=821) | 15% | 30% | 56% |
| | General exercise, for example, walking or swimming, is safe for everybody to do (n=820) | 13% | 16% | 71% |
| | Exercise works just as well for everybody, regardless of the amount of pain they have (n=823) | 49% | 29% | 22% |
| The effectiveness of exercise is independent of the presence or severity of radiographic findings | Exercise is effective for patients if an X-ray shows severe knee osteoarthritis (n=822) | 16% | 32% | 52% |
| Improvements in muscle strength and proprioception gained from exercise programmes may reduce the progression of knee and hip OA | Increasing the strength of the muscles around the knee stops the knee problem getting worse (n=824) | 16% | 29% | 55% |
| | Increasing the overall activity levels stops the knee problem getting worse (n=822) | 19% | 38% | 43% |
| **Items relating to the delivery of, and adherence to, exercise (number of respondents)** | | | | |
| Exercise therapy for OA of the hip or knee should be individualised and patient-centred taking into account factors such as age, comorbidity and overall mobility | Exercise for CKP is most beneficial when it is tailored to meet individual patient needs (n=823) | 1% | 9% | 90% |
| | A standard set of exercises is sufficient for every patient with chronic knee problems (n=821) | 51% | 36% | 13% |
| To be effective, exercise programmes should include… advice and education to promote a positive lifestyle change with an increase in physical activity | GPs should educate patients with CKP about how to change their lifestyle for the better (n=823) | 1% | 6% | 93% |
| | It is important that people with CKP increase their overall activity levels (n=824) | 1% | 10% | 89% |
| Adherence is the principal predictor of long-term outcome from exercise in patients with knee or hip OA | How well a patient complies with their exercise programme determines how effective it will be (n=825) | 3% | 11% | 86% |
| Strategies to improve and maintain adherence should be adopted, for example, long-term monitoring/review and inclusion of spouse/family in exercise | GPs should follow-up patients to monitor extent of continuation of exercises (n=823) | 30% | 37% | 34% |
| | It is the patient's own responsibility to continue doing their exercise programme (n=826) | 1% | 6% | 93% |

Consensus categorised according to: unanimity=100%, consensus=75%–99%, majority view=51–74%, no consensus=0%–50% (19,27).
Maximum missing data for any item were 2%.
CKP, chronic knee pain; GP, general practitioner, OA, osteoarthritis.

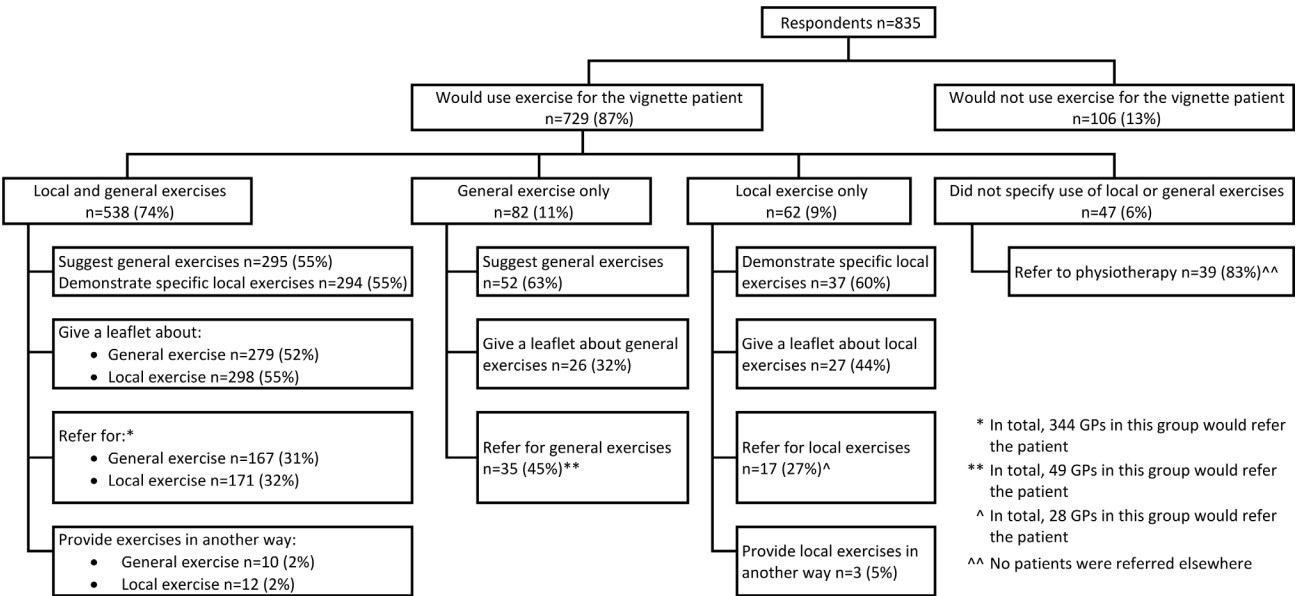

**Figure 1**  Flow chart summarising the exercise types and initiation methods used by general practitioners (GPs) for the vignette patient with chronic knee pain (CKP).

recommendations for OA have been published; thus familiarity with best-evidence recommendations may have increased.

GPs' reported behaviour was consistent with higher estimates from other physician questionnaire studies; between 9%–89% of GPs advising exercise[5 29–31] and 10%–77% referring patients with CKP to physiotherapy.[5 29–31] Given that estimates of GPs' exercise use are similar internationally, for example, advice to exercise was 46%–76% in the UK[32 33] and 12%–59% in the USA,[31 34–37] the results are likely to be relevant beyond the UK. While the most common general exercise suggestions (swimming, walking and cycling) were clinically appropriate, the extent to which this advice would be translated into action may be questionable from previous literature. Walking is acceptable to patients[38] and it improves function,[39] while it is also low impact, easily accessible,[39] adaptable to patient preferences and easy to incorporate into everyday life. However, it is unclear how acceptable or realistic it is for patients with CKP to engage in cycling or swimming.[40 41] For example, having to pay (eg, for equipment or instructors) is a recognised barrier to physical activity engagement.[13 42] Combining the findings of uncertain appropriateness of (some) suggested modes of exercise with only 5% of GPs using general exercise explicitly stating that their advice should be tailored to their patients' interests and abilities, suggests a need for greater focus on individualising exercise. As many GPs reported having insufficient time and expertise to use exercise, referral to other HCPs may be appropriate. However physiotherapists report similar uncertainties about the safety and efficacy of exercise[19] and suboptimal use of both exercise types, favouring local more than general exercise approaches.[43] Further, it is unlikely to be an economically viable approach for GPs to manage all patients with CKP by referring them to physiotherapists. It is possible therefore, that solely relying on physiotherapists to deliver exercise interventions may not ensure all patients receive tailored, specific instruction on both exercise types. Finally, a third of respondents

**Table 4**  Methods used to initiate local and general exercise by general practitioners (GPs) using both exercise types

| Methods used to include general exercises | Methods used to include local exercises | | | |
|---|---|---|---|---|
| | Does not demonstrate, give leaflet nor refer | Refers and/or demonstrates | Leaflet only | Leaflet and demonstrates, and/or refer |
| *Does not suggest, give leaflet nor refer* | 0% | 1% | 0% | <1% |
| Refers and/or suggests only | <1% | 33% | 6% | 7% |
| *Leaflet only* | 0% | 4% | 11% | 7% |
| Leaflet, suggests and/or refers | <1% | 6% | 6% | 17% |

Management strategies used by >5% GPs are emboldened. The responses in the box are those that are consistent with evidence-based recommendations (ie, providing advice and written information about both types of exercise and/or referring if needed). n=535

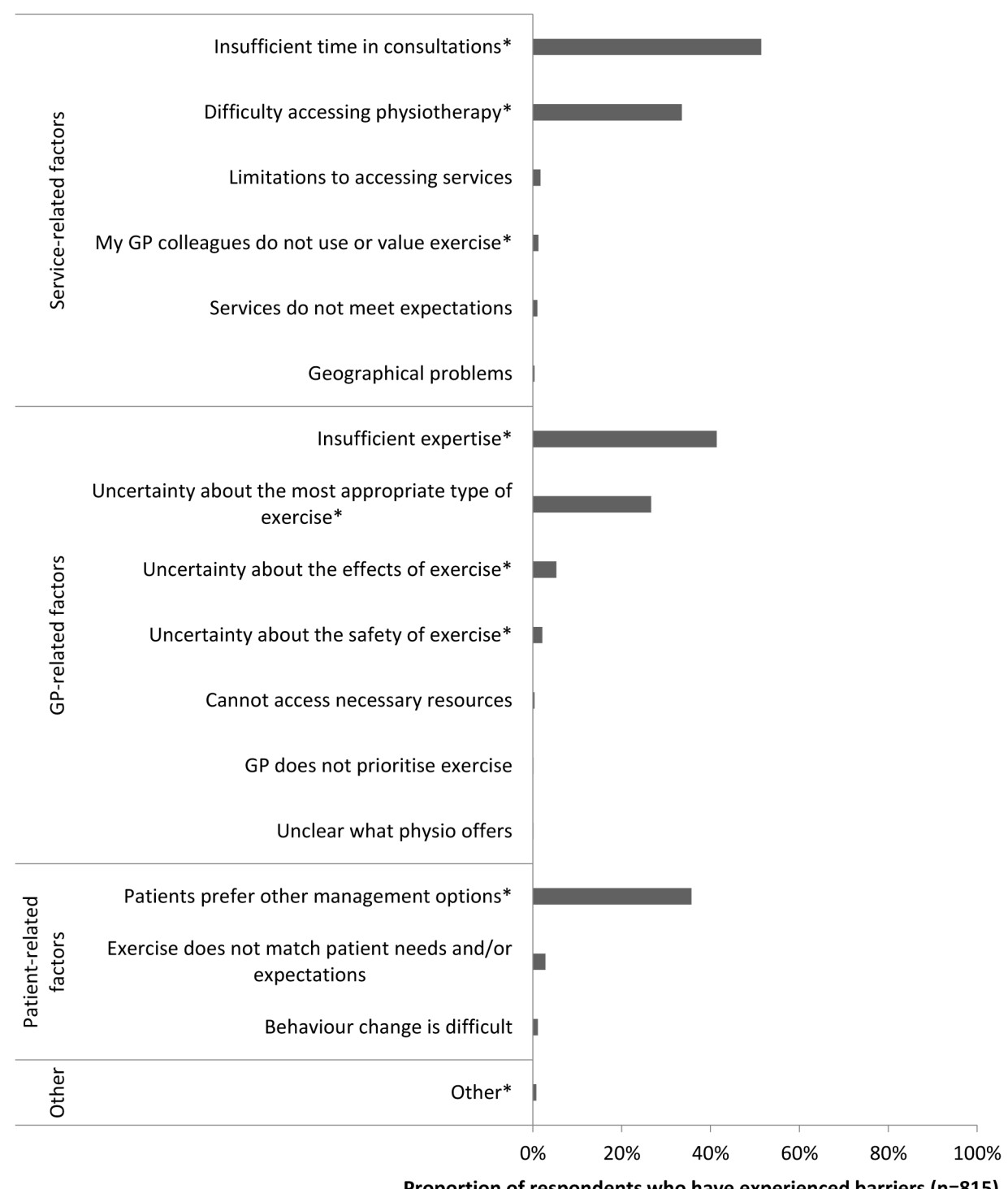

Figure 2  Barriers to using exercise reported by (general practitioners) GPs.

reported that they would opportunistically follow-up the vignette patient to check adherence to exercise. However, previous research examining the consultation behaviour of patients with CKP has shown that while many patients may consult again with other health problems, CKP is often not recorded again.[44] Thus the GPs' reported intentions regarding follow-up are likely to be unrealistically optimistic.

### Strengths and limitations
This large UK survey is the first known to directly, specifically and concurrently investigate the attitudes, beliefs

and behaviours of GPs regarding exercise for patients with CKP. The questionnaire was pretested and piloted before being used in this main study. Employing a vignette provided a consistent patient scenario to all GPs[45–48] and minimised the confounders inherent in observational research investigating clinical management using real patients.[46 47] Limitations of this study include the likely overestimation of exercise use among GPs given the low response rate (response bias), the self-report nature of the questionnaire (social desirability bias) and the relatively uncomplicated vignette case. Due to the use of survey methodology, we could not explore the reasons underlying GPs' responses. However, as GPs should be using exercise for all patients with CKP, the results of this survey are valuable for indicating an apparent evidence-practice gap in the way in which GPs employ exercise with this patient group.

### Implications for future practice

Implementation of current evidence-based recommendations for the management of CKP needs to be improved among GPs in the UK. It is possible that at least part of the problem in implementing the recommendations is the lack of explicit guidance regarding the role of GPs, and other members of the multidisciplinary team who may be involved in managing patients with this condition. This is perhaps reflected by the variable perceptions among GPs regarding their role in managing CKP and the extent to which they believe they should provide exercise advice or prescription. However, given the apparent association between perceived role and behaviour among the GPs who responded to the survey (reported elsewhere)[17], greater clarity of roles and expectations of all professional groups would be a good starting point for improving implementation of guidelines.

To deliver best practice for patients with CKP, strategies which target both GPs and the wider primary care team are needed. Two key areas should be addressed: (1) development of a pragmatic approach for GPs to initiate individualised local and general exercise and (2) identification of additional methods of initiating exercise and/or supporting patients to continue with exercise that do not solely rely on GPs. Given that theoretical patient behaviour change models[49] often involve a balance of perceived value and risks/burdens of undertaking the new behaviour, a pragmatic approach for GPs initiating exercise among patients with CKP would need to highlight the value of exercise to patients, its role relative to other interventions, and practical ways to undertake specific and individualised exercise. Such a best practice approach was suggested by Khan *et al*[50] who recommended that GPs should encourage the use of exercise (eg, by asking about physical activity at each consultation), consider the '5 A's' of physical activity counselling (assess, advise, agree, assist and arrange),[51] write an exercise prescription,[52–54] and refer or signpost to appropriate professionals or resources for exercise support and/or follow-up. Supplementary written leaflets or 'guidebooks' seem to be acceptable and useful[55] and can be accessed or signposted within consultations; for example, the Keele University OA guidebook[56] and the Arthritis Research UK knee OA booklet.[57] Given that patients with CKP commonly have comorbidities[38 58 59] and multiple joint pain, GPs could use this opportunity to detect and manage comorbid conditions that may directly impact the use of exercise (eg, cardiovascular disease, depression)[60], to relay the synergistic benefits of exercise for all relevant morbidities, and to make explicit that CKP should not prevent exercise for other conditions. Practical methods to help GPs provide the above information, in the limited time available, need to be developed and may include personalised written care plans.[61] Given that time was the most frequently reported barrier to GPs initiating exercise, service delivery models may need to change such that exercise initiation, support and/or follow-up is primarily undertaken by other professionals such as physiotherapists,[62 63] practice nurses,[64] health trainers or local gym personnel. Direct access may enhance patients' utilisation of physiotherapists[65] and alternative, low GP burden, strategies could be explored to promote exercise use and/or follow-up. For example, technology enabled care services[66] have shown promise when used to support exercise interventions among patients with cardiac[67] and chronic lung disease.[68]

## CONCLUSIONS

Although the majority of UK GPs who responded to the questionnaire survey were positive about exercise for patients with CKP and used exercise in their clinical management, this survey identified GP uncertainties with respect to the safety and efficacy of exercise and suboptimal approaches to the initiation of exercise with patients. GPs' use of exercise may be improved by addressing the key barriers of time and expertise, by developing a pragmatic approach that supports GPs to initiate individualised exercise with patients, and/or by other professionals taking on this role.

**Acknowledgements** The authors thank the Keele Clinical Trials Unit for support to this project. The authors also thank the staff within the Research Institute for Primary Care and Health Sciences who supported the study and the GPs who participated.

**Contributors** EC, NEF, MP, TR and ER participated in the design of the study, analysis of the results and helped to draft the manuscript. All authors read and approved the final manuscript.

**Funding** This paper presents independent research funded by the Arthritis Research UK Centre in Primary Care grant (Grant Number 20202). NEF, an NIHR Senior Investigator, is supported through an NIHR Research Professorship (NIHR-RP-011-015). EC was funded by an NIHR Academic Clinical Fellowship and subsequently by the NIHR School for Primary Care Research bridging funds. The views expressed in this publication are those of the authors and not necessarily those of the NHS, the NIHR or the Department of Health.

**Competing interests** None declared.

**Patient consent** Detail has been removed from these case descriptions to ensure anonymity. The editors and reviewers have seen the detailed information available and are satisfied that the information backs up the case the authors are making.

**Ethics approval** Keele University (UK) Ethical Review Panel.

**Provenance and peer review** Not commissioned; externally peer reviewed.

**Data sharing statement** The data sets analysed during the current study are available from the corresponding author on reasonable request.

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
