## [Reviewer comments · BMJ Open]

ARTICLE DETAILS

TITLE (PROVISIONAL)	GP attitudes, beliefs and behaviours regarding exercise for chronic knee pain: a questionnaire survey
AUTHORS	Cottrell, Elizabeth; Foster, Nadine; Porcheret, Mark; Rathod, Trishna; Roddy, Edward

VERSION 1 - REVIEW

REVIEWER	Thorlene Egerton The University of Melbourne Australia Nadine Foster, second author, is a CI on the Centre for Research Excellence which pays my salary. She is a current collaborator on projects with others in my organisation although I have not directly collaborated with her on any projects, nor do I have any planned in the future.
REVIEW RETURNED	28-Nov-2016

GENERAL COMMENTS	Comments for the authors The paper is well written and interesting, and I agree there is a need for this research question to be addressed. I do have some comments that I hope will help improve the clarity and usefulness of the data collected and the article. 1. In the first part of the Results, I think it would be helpful to provide the 'correct' response. At the moment the beliefs are reported but the reader is not signposted to the response the investigators would consider correct. This might be done by providing the relevant NICE recommendation or what the evidence currently points to. Without the 'correct' answer as a reference the results are difficult to interpret. For example, in relation to the whether exercise can stop the knee problem from getting worse – does the evidence suggest it can? Should the GPs be expected to answer 'agree' or 'disagree' to this question? In my opinion, it depends on what the 'problem' is defined as. If the GP interprets the 'problem' as pain and other symptoms, then I would expect them to answer 'agree'. However, if they interpret the 'problem' as structural damage, then they should probably answer 'disagree'. A second example is in relation to whether exercise works just as well for everybody regardless of pain. In my opinion
--

the answer to this question also depends on the interpretation. If the GP interprets the question as 'Does exercise work for everybody?', then I think they could be expected to 'agree', however if they focus on 'equally well for everybody' then surely it would be reasonable to 'disagree'. What do the authors think is the 'correct' answer and why. Are 49% correct or 49% incorrect?

A third example is in relation to who has the responsibility for adherence. What do the authors suggest is the 'correct' answer? One could argue that the GP has an important role in facilitating adherence or that ultimately only up to the patient?

Finally, should GPs prescribe? The answer again depends on the interpretation of the question. I assume there should be agreement with the general principle that GPs should recommend exercise, but some GPs might argue that they shouldn't be 'prescribing' but rather advising and negotiating. They might also argue that prescribing exercise is not needed for patients already exercising appropriately.

It is unclear to me how the data to these questions should be interpreted and some further guidance for the reader would be valuable.

2. I am not sure how useful data from the question related to 'increasing muscle strength / overall activity stops the knee problem getting worse'. They are difficult questions to answer because to my knowledge there is no evidence that exercise or activity will stop worsening of either knee pain or structural damage so the correct answer should surely be to disagree, however 55% and 43% have agreed. Again, some explanation of what the correct answer is and why would be very valuable.
3. In methods, it is stated "Free text responses underwent thematic analysis". The only free text answers reported in results, I think, were the types of activity (swimming, cycling, and patient preference). I'm not sure that thematic analysis was used. I suggest in the Methods to report in a bit more detail which of the free text fields in the questionnaire were used for this study and how they were used.
4. In general, it is not clear whether the study is based on principles that GPs should be prescribing or recommending specific exercise or activity, or whether they should be following behaviour change principles and providing

individualised education and advice and supporting patients to choose their own activity and exercises. There seems to be some inconsistency within the manuscript. Some questions were worded as prescribe with the implication that they should prescribe, and then the Discussion on lines 167-69 suggests that GPs should not be that specific in their advice. Please clarify with regard what the NICE guidelines recommend and check for consistency throughout the manuscript and therefore also indicate what the ideal responses are.

5. Strengths. The vignette was only used for one section of the questionnaire and therefore it is not entirely accurate to suggest that the use of the vignette helped 'minimise the confounders' as there is clearly confounders and possible variation in interpretation of the questions that will affect the answers to questions in other sections of the survey.

6. What were the differences between GPs with/without personal experience of knee pain or with/without relevant post graduate training?

7. The Table 4 is quite difficult to interpret. I wonder if, since you suggest that either demonstration or referral is consistent with good practice, demonstration and/or referral can be collapsed together, and suggest and/or refer can be collapsed together. That would remove three columns and three rows, and there would only be one cell that is 'consistent with recommendations' instead of 9 cells.

8. The first paragraph of Discussion is given the title 'Summary', but I don't think it is a good overall summary. Consider re-writing the first paragraph to summarise the research question and all the main findings.

9. Line 238. Start a new paragraph for the new topic.

10. I am not sure the value of the discussion about the appropriateness of swimming and cycling in this paper since there was limited information about the patient preferences in the vignette and the study was more about GPs beliefs and behaviours than a critique on the different types of physical activity. Again it is unclear to me what the 'correct'

	response is to this question. 11. I am not sure the value of the discussion starting on Line 241 as elsewhere you have suggested that referral to other professionals is consistent with recommended practice. 12. I am not sure the value of the discussion starting on Line 246 as you are basically saying your data is not reliable. 13. In the implications section, since you previously pointed out that patients often do not believe in the effectiveness and appropriateness of exercise/activity, and that a barrier to GP delivering exercise advice is patient expectations, I suggest you add patients as targets for 'strategies' such as education as well. 14. I think there is scope for further discussion around the results especially in relation to possible variations in interpretation of the some of the questions.
--	---

REVIEWER	Jesper Knoop Amsterdam Rehabilitation Research Center; Reade, center for rehabilitation and rheumatology (Amsterdam, the Netherlands)
REVIEW RETURNED	17-Jan-2017

GENERAL COMMENTS	This is a highly relevant and well-written manuscript about the GP's attitudes, beliefs and behaviors about knee pain/knee OA. Only a few comments/suggestions for further improvement from my side: Introduction: - please add how exercise/exercise therapy is recommended in GP-guidelines in the UK Statistic analysis/Results: - I would think that 'expertise in OA' is a major confounder in the attitudes, beliefs and attitudes of GP's, which seems to be confirmed with your finding that 'insufficient experience' is top-2 barrier to use exercise. Is it possible to adjust your analysis for this potential confounder, or split your results between 'low experience' and 'high experience'? Results: - It would be highly relevant if information could be added on GP's beliefs about the reasons that exercise may not be as effective for each patient. Has this been collected in your questionnaire? This information - if available - should also be discussed in the
--

	Discussion.  - what are the reasons for GP's not to participate in your study? - Table 4 is very difficult to read; I would suggest to remove this table. Discussion:  - I would suggest to add the following implication of your findings, namely that guideline implementation may need to be enhanced for GP's in the UK, based on your findings.
--	---

VERSION 1 – AUTHOR RESPONSE

Reviewer 1: Thorlene Egerton

REVIEWER 1 POINT 1. In the first part of the Results, I think it would be helpful to provide the 'correct' response. At the moment the beliefs are reported but the reader is not signposted to the response the investigators would consider correct. This might be done by providing the relevant NICE recommendation or what the evidence currently points to. Without the 'correct' answer as a reference the results are difficult to interpret. For example:

- a) in relation to the whether exercise can stop the knee problem from getting worse – does the evidence suggest it can? Should the GPs be expected to answer 'agree' or 'disagree' to this question? In my opinion, it depends on what the 'problem' is defined as. If the GP interprets the 'problem' as pain and other symptoms, then I would expect them to answer 'agree'. However, if they interpret the 'problem' as structural damage, then they should probably answer 'disagree'.
- b) A second example is in relation to whether exercise works just as well for everybody regardless of pain. In my opinion the answer to this question also depends on the interpretation. If the GP interprets the question as 'Does exercise work for everybody?', then I think they could be expected to 'agree', however if they focus on 'equally well for everybody' then surely it would be reasonable to 'disagree'. What do the authors think is the 'correct' answer and why. Are 49% correct or 49% incorrect?
- c) A third example is in relation to who has the responsibility for adherence. What do the authors suggest is the 'correct' answer? One could argue that the GP has an important role in facilitating adherence or that ultimately only up to the patient?
- d) Finally, should GPs prescribe? The answer again depends on the interpretation of the question. I assume there should be agreement with the general principle that GPs should recommend exercise, but some GPs might argue that they shouldn't be 'prescribing' but rather advising and negotiating. They might also argue that prescribing exercise is not needed for patients already exercising appropriately.
- e) It is unclear to me how the data to these questions should be interpreted and some further guidance for the reader would be valuable.

OUR RESPONSE: The MOVE Consensus recommendations (Roddy et al, cited in the paper), upon which the attitude statements were based, have been inserted into Table 3 to clarify these issues. These are statements that have been used previously among physiotherapists by Holden et al, cited in the paper, and they are based on consensus informed by best available evidence. To some extent the answers to the questions raised by the reviewer are unknown, and recent OA guidelines do not address all of these statements nor make recommendations about them. Thus there is no clearly 'right' or 'wrong' answer to the questions we asked GPs in this study, rather our study was designed to assess GPs' attitudes and beliefs in order to reach some understanding of what they think they should be doing with respect to exercise for chronic knee pain in older adults. This point has now been made more explicitly in the introduction on page 5 "While it is recognised that delivery of care for CKP is multidisciplinary, the exact roles and explicit expectations of GPs (and other professionals) regarding the delivery of core management approaches is not provided within current guidelines. This could have the consequence that no professional undertakes certain activities in the belief that others will." and page 6 "The role that GPs perceive themselves to have in delivering these management

approaches was also not clear”.

With specific reference to point (a) above, the GPs were provided with a definition of chronic knee pain (CKP) in the questionnaire (provided with the manuscript as a supplementary file) “We are interested in your clinical opinion about patients aged 45 years and over with chronic knee pain. In this age group chronic knee pain is almost always due to knee osteoarthritis. Please answer all of the following questions using the definition of chronic knee pain as follows: knee pain and associated symptoms that have been present for more than 3 months not resulting from a fracture, infection, systemic rheumatological problem, metastases or surgery.”

REVIEWER 1 POINT 2. I am not sure how useful data from the question related to ‘increasing muscle strength/overall activity stops the knee problem getting worse’. They are difficult questions to answer because to my knowledge there is no evidence that exercise or activity will stop worsening of either knee pain or structural damage so the correct answer should surely be to disagree, however 55% and 43% have agreed. Again, some explanation of what the correct answer is and why would be very valuable.

OUR RESPONSE: Thank you for this point. Please see the additional column in table 3 on pages 11-12 that provides the associated MOVE consensus recommendations.

REVIEWER 1 POINT 3. In methods, it is stated “Free text responses underwent thematic analysis”. The only free text answers reported in results, I think, were the types of activity (swimming, cycling, and patient preference). I’m not sure that thematic analysis was used. I suggest in the Methods to report in a bit more detail which of the free text fields in the questionnaire were used for this study and how they were used.

OUR RESPONSE: This section of the methods has been amended for clarity to “Responses to the GP attitude statements were condensed into three categories ((strongly) disagree, neither disagree nor agree, and (strongly) agree) and free-text responses (associated “other” responses and regarding type of exercise the GPs would use) underwent thematic analysis, categorising responses into pre-defined categories that emerged from responses to the pilot study and developing new categories as appropriate, before commencing descriptive analyses.”

REVIEWER 1 POINT 4. In general, it is not clear whether the study is based on principles that GPs should be prescribing or recommending specific exercise or activity, or whether they should be following behaviour change principles and providing individualised education and advice and supporting patients to choose their own activity and exercises. There seems to be some inconsistency within the manuscript. Some questions were worded as prescribe with the implication that they should prescribe, and then the Discussion on lines 167-69 suggests that GPs should not be that specific in their advice. Please clarify with regard what the NICE guidelines recommend and check for consistency throughout the manuscript and therefore also indicate what the ideal responses are.

OUR RESPONSE: It was not our intention to suggest that GPs should be prescribing exercise, indeed, we do not feel that GPs will all be confident and capable to do so. There is no clear recommendation within the original (2008) or the revised (2014) NICE guidelines about exactly what GPs should be doing and this was the reason for investigating GP perceptions. Additions to the introduction section in response to point 1 help to clarify this and an additional paragraph has been added to the implications on page 17 to address both this and Reviewer 2’s sixth point, as follows: “Implementation of current evidence-based recommendations for the management of CKP needs to be improved among GPs in the UK. It is possible that at least part of the problem in implementing the recommendations is the lack of explicit guidance regarding the role of GPs, and other members of the multidisciplinary team who may be involved in, managing patients with this condition. This is perhaps reflected by the variable perceptions among GPs regarding their role in managing CKP and the extent to which they believe they should provide exercise advice or prescription. However, given the apparent association between perceived role and behaviour among the GPs who responded to the survey (reported elsewhere [17]), greater clarity of roles and expectations of all professional groups

would be a good starting point for improving implementation of guidelines.”

REVIEWER 1 POINT 5. Strengths. The vignette was only used for one section of the questionnaire and therefore it is not entirely accurate to suggest that the use of the vignette helped ‘minimise the confounders’ as there is clearly confounders and possible variation in interpretation of the questions that will affect the answers to questions in other sections of the survey.

OUR RESPONSE: With regards to confounders and variation in interpretation, we agree that this is an issue however this was mitigated by undertaking a pilot survey (Cottrell et al 2015, cited in the paper) which tested out the questionnaire prior to this main survey. A sentence emphasising this has been added to the strengths and limitations on page 17 of the manuscript: “The questionnaire was pre-tested and piloted before being used in this main study.” The reviewer is correct in that the vignette was only used in one section of the questionnaire, however the comment regarding reducing confounding was in relation to investigating clinical management of CKP. We have therefore explicitly mentioned this in the strengths and limitations section of the discussion, in order to clarify this point – the new sentence now reads “Employing a vignette provided a consistent patient scenario to all GPs[45-48] and minimised the confounders inherent in observational research investigating clinical management using real patients.[46,47]”

REVIEWER 1 POINT 6. What were the differences between GPs with/without personal experience of knee pain or with/without relevant post graduate training?

OUR RESPONSE: The results of these analyses (that personal experience of CKP and postgraduate training were not associated with differences in exercise use) have been published in a separate paper: Cottrell E, Roddy E, Rathod T, Porcheret M, Foster NE. What influences general practitioners’ use of exercise for patients with chronic knee pain? Results from a national survey. BMC Family Practice. 2016; 17: 172. As these findings have been reported elsewhere they have not been included in this paper however a sentence stating this (and citing Cottrell et al 2016 in the paper) has been included at the end of the introduction on page 6: “Analysis of factors associated with the use of exercise among this group have been published elsewhere.[17]”.

REVIEWER 1 POINT 7. The Table 4 is quite difficult to interpret. I wonder if, since you suggest that either demonstration or referral is consistent with good practice, demonstration and/or referral can be collapsed together, and suggest and/or refer can be collapsed together. That would remove three columns and three rows, and there would only be one cell that is ‘consistent with recommendations’ instead of 9 cells.

OUR RESPONSE: Thank you for this observation - Table 4 on page 13 has been condensed, for clarity.

REVIEWER 1 POINT 8. The first paragraph of Discussion is given the title ‘Summary’, but I don’t think it is a good overall summary. Consider re-writing the first paragraph to summarise the research question and all the main findings.

OUR RESPONSE: The research question has been restated at the start of the discussion on page 14 to orient the reader to the summary. We have summarised the key findings in this first paragraph.

REVIEWER 1 POINT 9. Line 238. Start a new paragraph for the new topic.

OUR RESPONSE: A paragraph break has been added under “comparison with existing literature”, separating the discussion about the appropriateness of exercise types from use of referral. On the submitted version, there is not a clear break on line 238.

REVIEWER 1 POINT 10. I am not sure the value of the discussion about the appropriateness of swimming and cycling in this paper since there was limited information about the patient preferences in the vignette and the study was more about GPs beliefs and behaviours than a critique on the different types of physical activity. Again it is unclear to me what the ‘correct’ response is to this

question.

OUR RESPONSE: This text was included as it demonstrates that even among the GPs who are behaving in line with guidelines and recommending general exercise, this may not be done in a way that patients find easy and acceptable to translate into action. This has been made more explicit within the text on page 16 in the discussion section, as follows: "While the most common general exercise suggestions (swimming, walking and cycling) were clinically appropriate, the extent to which this advice would be translated into action may be questionable from previous literature." Similar to our response to your point 1 above, there is no "correct" response, however we have commented that "...it is unclear how acceptable or realistic it is for patients with CKP to engage in cycling or swimming.[39,40]"

REVIEWER 1 POINT 11. I am not sure the value of the discussion starting on Line 241 as elsewhere you have suggested that referral to other professionals is consistent with recommended practice.

OUR RESPONSE: We recognise the dissonance regarding referrals highlighted by the reviewer. We had already highlighted that referral may be appropriate, however we have amended this section on page 16 to improve clarity, as follows: "However physiotherapists report similar uncertainties about the safety and efficacy of exercise[19] and suboptimal use of both exercise types, favouring local more than general exercise approaches.[43] Further, it is unlikely to be an economically viable approach for GPs to manage all patients with CKP by referring them to physiotherapists. It is possible therefore, that solely relying on physiotherapists to deliver exercise interventions may not ensure all patients receive tailored, specific instruction on both exercise types."

Reviewer 2: Jesper Knoop

REVIEWER 2 POINT 1. Introduction: - please add how exercise/exercise therapy is recommended in GP-guidelines in the UK

OUR RESPONSE: This content was present in the background section to some extent, however it has been clarified further by adding in detail of the relevant guideline on page 5, as follows: "In line with wider self-management strategies, best practice outlined by the National Institute for Health and Care Excellence (NICE) OA guidelines with regards to integrating exercise into the management of patients with CKP involves providing verbal advice about both general and local exercise (which should be specific and individualised[4,13]) supported with written information.[4]"

REVIEWER 2 POINT 2. Statistic analysis/Results: - I would think that 'expertise in OA' is a major confounder in the attitudes, beliefs and attitudes of GP's, which seems to be confirmed with your finding that 'insufficient experience' is top-2 barrier to use exercise. Is it possible to adjust your analysis for this potential confounder, or split your results between 'low experience' and 'high experience'?

OUR RESPONSE: We did not collect data on "expertise in OA" explicitly, only with regards to self-reported postgraduate training (which involved CKP), having a special interest in musculoskeletal conditions and personal experience of CKP. We are therefore unable to split results between those with "low experience" and "high experience". However, this point is aligned to reviewer 1's sixth point, and, as stated there, results of the relevant analyses have been published in a separate paper: Cottrell E, Roddy E, Rathod T, Porcheret M, Foster NE. What influences general practitioners' use of exercise for patients with chronic knee pain? Results from a national survey. BMC Family Practice. 2016; 17: 172. While personal experience of CKP and postgraduate training were not associated with differences in exercise use, all 50 GPs who reported they had a special interest in musculoskeletal conditions reported that they would use exercise (compared with 87% of GPs who did not report this special interest). As these findings have been reported elsewhere they have not been included in this paper however a sentence stating this (and citing the paper) has been included at the end of the introduction section on page 6.

REVIEWER 2 POINT 3. Results: It would be highly relevant if information could be added on GP's

beliefs about the reasons that exercise may not be as effective for each patient. Has this been collected in your questionnaire? This information - if available - should also be discussed in the Discussion.

OUR RESPONSE: While we appreciate that this would be interesting to know, we did not collect this level of detail in the questionnaire survey (provided as a supplementary file in the original submission). This has been added to the limitations of the paper on page 17: "Due to the use of survey methodology, we could not explore the reasons underlying GPs responses."

REVIEWER 2 POINT 4. Results: What are the reasons for GP's not to participate in your study?

OUR RESPONSE: Within the methods section, we have added text indicating that in addition to a minimum data set (MDS) being requested, the reason for return of MDS (rather than full response) was collected. Within "response" under "results" we have added on page 8 "The most common reason for returning MDS, rather than a full response was, having insufficient time (n=408, 87%)." We do not know the reasons why GPs did not respond in any way to our survey (those who did not respond to the main survey or to the MDS).

REVIEWER 2 POINT 5. Results: Table 4 is very difficult to read; I would suggest to remove this table.

OUR RESPONSE: We appreciate the difficulty with this table, therefore, in line with Reviewer 1's comments we have condensed it to simplify it rather than remove it completely. We hope this is acceptable.

REVIEWER 2 POINT 6. Discussion: - I would suggest to add the following implication of your findings, namely that guideline implementation may need to be enhanced for GP's in the UK, based on your findings.

OUR RESPONSE: Thank you for this suggestion. We have added a section to the implications on page 17; see response to Reviewer 1 point 4.

VERSION 2 – REVIEW

REVIEWER	Dr. J. Knoop Reade, the Netherlands
REVIEW RETURNED	28-Mar-2017
GENERAL COMMENTS	No further comments as the manuscript has been adequately changed based on the reviewers comments.